# Non-Coding RNAs: Novel Players in Insulin Resistance and Related Diseases

**DOI:** 10.3390/ijms22147716

**Published:** 2021-07-19

**Authors:** Caterina Formichi, Laura Nigi, Giuseppina Emanuela Grieco, Carla Maccora, Daniela Fignani, Noemi Brusco, Giada Licata, Guido Sebastiani, Francesco Dotta

**Affiliations:** 1Diabetes Unit, Department of Medicine, Surgery and Neurosciences, University of Siena, 53100 Siena, Italy; catefo@libero.it (C.F.); launigi@gmail.com (L.N.); giusy.grieco.90@gmail.com (G.E.G.); dfignani@gmail.com (D.F.); noemibrusco91@gmail.com (N.B.); giadalicata.92@gmail.com (G.L.); sebastianiguido@gmail.com (G.S.); 2Fondazione Umberto Di Mario, c/o Toscana Life Sciences, 53100 Siena, Italy; 3Section of Medical Pathophysiology, Food Science and Endocrinology, Department of Experimental Medicine, Sapienza University, 00185 Rome, Italy; carla.maccora@gmail.com; 4Tuscany Centre for Precision Medicine (CReMeP), 53100 Siena, Italy

**Keywords:** insulin resistance, non-coding RNAs, obesity, diabetes, fatty liver disease

## Abstract

The rising prevalence of metabolic diseases related to insulin resistance (IR) have stressed the urgent need of accurate and applicable tools for early diagnosis and treatment. In the last decade, non-coding RNAs (ncRNAs) have gained growing interest because of their potential role in IR modulation. NcRNAs are variable-length transcripts which are not translated into proteins but are involved in gene expression regulation. Thanks to their stability and easy detection in biological fluids, ncRNAs have been investigated as promising diagnostic and therapeutic markers in metabolic diseases, such as type 2 diabetes mellitus (T2D), obesity and non-alcoholic fatty liver disease (NAFLD). Here we review the emerging role of ncRNAs in the development of IR and related diseases such as obesity, T2D and NAFLD, and summarize current evidence concerning their potential clinical application.

## 1. Introduction

NcRNAs represent approximately 98% [1] of the transcriptional production of the human genome and are generally not translated into proteins [2]. NcRNAs include long non-coding RNAs (lcnRNAs), microRNAs (miRNAs), piwi-interacting RNAs (piRNAs), ribosomal RNAs (rRNAs), small nuclear RNAs (snRNAs), small nucleolar RNAs (snoRNAs) and transfer RNAs (tRNAs) [3]. According to transcript length, ncRNAs can be classified into two categories: small ncRNAs up to 200 ribonucleotides in length (best represented by microRNAs, but including also snRNAs, snoRNAs and piRNAs) and long ncRNAs over 200 nucleotides [4]. Based on their biological function, ncRNAs can be classified into infrastructural and regulatory types. Infrastructural ncRNAs include ribosomal RNAs (rRNAs), small nuclear RNAs (snRNAs) and transfer RNAs (tRNAs) while regulatory ncRNAs are mainly represented by circular RNAs (circRNAs), long non coding RNAs (lncRNAs), microRNAs (miRNAs) and piwi-interacting RNAs (piRNAs) [5]. Although they do not encode proteins, ncRNAs are functionally active and contribute to the regulation of protein-coding gene expression [6] through different mechanisms including modification of chromatin structure, repressing/activating transcription and post-transcriptional regulation [3,7]. These RNA molecules seem to regulate key developmental processes and homeostasis, metabolism, cell differentiation and growth [3,8]. A growing body of evidence indicates that altered expression of ncRNAs patterns (e.g., due to mutations or dysregulation) is related to the development and evolution of several diseases, including metabolic ones [7,9,10]. Major metabolic diseases such as obesity, diabetes mellitus and NAFLD, along with metabolic syndrome (MetS) and dyslipidemia, have reached epidemic proportions worldwide in the past few decades [11,12,13] with deleterious consequences, including increased morbidity and mortality [14]; hence, the necessity for a better understanding of their underlying pathophysiology.

Obesity is a multifactorial chronic disease, characterized by an imbalance between energy intake and energy expenditure with subsequent excessive fat accumulation. Obesity is caused by multiple factors interaction, such as food intake, physical inactivity, genetic and epigenetic predisposition, environmental factors and nutritional components [15,16]. This condition, related to white adipose tissue dysfunction, represents an important risk factor for many diseases, including cancer, cardiovascular disorders, diabetes and MetS [17]. T2D is also a complex multi-factorial disease, caused by a progressive loss of adequate insulin secretion by β-cell, resulting in hyperglycemia. T2D commonly develops on the background of insulin resistance (IR) and involves genetic, epigenetic, and environmental factors [18]. Most patients with T2D are overweight/obese, mainly with abdominal fat deposition, responsible of some degree of IR [18]. NAFLD is characterized by lipid accumulation in >5% of hepatocytes (as determined by liver histology), in the absence of other causes, such as autoimmunity, drug and alcohol abuse or viral hepatitis [19,20]. NAFLD is the most common liver disease in western countries [21]. Prevalence of NAFLD is rising in parallel to a worldwide increase in diabetes and MetS [22,23], and it is estimated to occur in up to 45% of the general population—but is even doubled in individuals with MetS [13]. The strong association of NAFLD with obesity and T2D is mainly attributable to IR, leading to visceral adiposity and lipid accumulation in the liver [20]. NAFLD is a clinically relevant and progressive disease, usually beginning as benign steatosis, but if not treated it can progress to nonalcoholic steatohepatitis (NASH—fatty liver with inflammation), fibrosis, and up to cirrhosis and hepatocellular carcinoma (HCC) in 10–25% of cases [13,21]. It is increasingly evident that NAFLD is a multisystem disease, affecting several extra-hepatic organs and involving different regulatory pathways. As a matter of fact, NAFLD increases T2D risk, cardiovascular diseases and chronic kidney disease [24]. Its pathogenesis implicates complex interactions between genetic predisposition and environmental risk factors including obesity, IR, dyslipidemia, diabetes and MetS [25,26]. Progression from steatosis to NASH is driven by different mechanisms, including lipotoxicity, oxidative stress and immune system activation. Even though extensively studied, the molecular mechanisms involved in steatosis development, as well as the pathways leading to progressive hepatocellular damage following lipid accumulation, are still poorly understood [23,26,27].

As already reported, one of the key underlying features of obesity, T2D and NAFLD is represented by IR, a pathological condition defined as the failure to coordinate glucose-lowering processes, i.e., suppression of gluconeogenesis, lipolysis, glycogen synthesis and cellular glucose uptake in response to insulin. The above-mentioned processes are the result of an impaired insulin signaling at the cellular level, in target tissues [28]. It is now well established that liver, as well as white adipose tissue (WAT) and skeletal muscle, plays a central role in maintaining this balance [29]. Pathological IR develops through complex interactions between genotype and lifestyle (e.g., lack of exercise and over-nutrition) [30]. However, much remains to be learned on the mechanisms that cause IR and the processes by which IR “promotes” diseases. Multiple molecular pathways contribute to the pathogenesis of metabolic disorders and their chronic complications. In particular, as mentioned above, they represent the result of a complex interaction among genetics, epigenetics, environmental and/or lifestyle factors [13]. Recently, the potential role of epigenetics in metabolic disease onset has been suggested [31,32]. NcRNAs have been suggested as major regulators of gene expression through epigenetic modifications in many processes, including inactivation of X chromatin [33], regulation of key metabolic genes function, cell cycle and cell differentiation control [34]. Over the last few years there has been a growing interest in studying ncRNAs, including microRNAs, lncRNAs and circular RNAs, which can act as regulators for epigenetic mechanisms [5,13,35]. More importantly, there is evidence of ncRNAs dysregulation in the regulatory pathways of lipid metabolism, in particular adipogenesis, adipocyte metabolism and hepatic lipid metabolism [36]. Moreover, ncRNAs seem to play an essential role in the IR modulation (particularly within the hepatic tissue) [37], as well as in the regulation of glucose homeostasis and of β-cell function [9,38]. Finally, several lncRNAs and microRNAs have been reported to be dysregulated in IR [37]. For these reasons, ncRNAs are regarded as promising novel biomarkers and therapeutic targets, owing to their regulatory functions [37]. In this review we describe the emerging role of ncRNAs in the development of IR and related diseases such as obesity, T2D and NAFLD.

## 2. Non-Coding RNAs Biogenesis and Function

### 2.1. Long Non-Coding RNAs

LncRNAs are defined as a group of heterogeneous ncRNAs, with sizes greater than 200 nucleotides in length, that cannot be translated into proteins [39]. LncRNAs actively contribute to the regulation of gene expression in multiple ways, so investigation on their biogenesis is important not only to differentiate them from other types of RNAs, but also to thoroughly understand their function in physiological and pathological conditions. The transcription of lncRNAs is often performed by RNA polymerase II from intergenic (lincRNAs), exonic or the distal protein-coding regions of the genome. The resulting pre-mature lncRNAs are 3′-polyadenylated and capped on the 5′-end with methyl-guanosine [40]. Subsequently, they can undergo alternative splicing in different manners: first, lncRNAs can interact with specific splicing factors; second, lncRNAs are able to form RNA-RNA duplexes with pre-mRNA molecules, and third, lncRNAs con influence chromatin remodeling, thus completing the splicing of target genes [41] (Figure 1). LncRNAs are classified based on their structure, function and localization. According to the most common classification, based on their position within the genome, lncRNAs are categorized in different subclasses: intergenic, located among two different genes that codify for proteins; intronic, located fully in intronic regions of protein-coding genes; bidirectional, located within 1 kb of the promoter region of protein-coding genes; sense, transcribed from the same strand and the same direction as the surrounding the codify genes; and antisense, transcribed from the opposite strand of surrounding protein-coding genes [42,43]. As for their mode of action, lncRNAs can affect gene regulation in three different ways: as competitors, by binding to DNA-binding proteins [44]; as recruiters, by recruiting epigenetic complexes, for example, during DNA methylation [45]; and finally, as precursors of small RNAs, especially microRNAs [46]. Based on their subcellular localization lncRNAs are classified into different groups: lncRNAs that accumulate and act in cis, once they are transcribed; those that can accumulate in cis once they are transcribed, but act in trans affecting genes located in a different location of the same chromosome or in another chromosomes; lncRNAs that localize in the nucleus in trans and act in trans, and lncRNAs released to the cytoplasm to carry out their roles. For instance, cytoplasmic lncRNAs can inhibit protein post-translational modifications, resulting into aberrant signal transduction [47,48]. Depending on their cellular function, lncRNAs can be divided into several categories: signal, decoy, guide and scaffold. Signal lncRNAs are localized in specific subcellular regions and respond to different stimuli at specific time points [49]. On the other hand, decoy lncRNAs regulate an effector by binding regulatory factors such as transcription factors and RNA-binding proteins [50]. Guide lncRNAs are implicated in directing the localization of ribonucleoprotein complexes to specific targets, thus regulating gene expression [49]. Finally, scaffold lncRNAs, are involved in structural roles, with reported effects on chromatin complexes and as histone modifiers [51]. Although many lncRNAs have been identified and their biogenesis and functions have been examined, the understanding of their biological roles is still under investigation.

### 2.2. MicroRNAs

MiRNAs are a class of small ncRNAs of a size range of 18–22 nt in length. MiRNAs are able to bind the 3’ untranslated region (3’UTR) of target mRNA, leading to its degeneration or suppressing its translation. Thus, this family of ncRNAs is implicated in gene expression regulation and in different biological processes. MiRNAs biogenesis, maturation, function and secretion are regulated by highly complex molecular mechanisms not yet fully elucidated [53]. RNA polymerase II transcribes a large part of miRNAs from their genes, generating primary miRNAs (pri-miRNAs). Pri-miRNAs are stem loop shaped RNA sequences, capped and polyadenylated and may also be spliced. Once processed, pri-miRNAs are recognized and cleaved, within the nucleus, by the multi-protein complex microprocessor [54,55]. Microprocessor complex is composed by two main molecules, the double-stranded RNase III enzyme DROSHA and the double-stranded RNA-binding protein DGCR8. DROSHA cleaves, by its RNase III domains, at two different points of the double strand RNA (dsRNA) towards the base of the stem-loop, generating a ~70 nucleotide hairpin-shaped precursor miRNA (pre-miRNA). This latter has an overhang at the 3′ end of 2 nucleotides left by the asymmetrical cut made by DROSHA. After generation, exportin-5 (XPO5)/RanGTP complex export pre-miRNAs to the cytoplasm [56,57], where they are additionally processed by DICER. The function of this RNase III enzyme is to generate duplexes in a size range of 22 nucleotides comprising a guide and a passenger strand. The guide strand, preferentially the most thermodynamically stable, is loaded into the argonaute family protein (AGO1-4 in humans) in an ATP-dependent manner, while the passenger strands are cleaved by AGO2 and degraded by cellular machinery [58,59] (Figure 1). On the other hand, there is evidence of non-canonical miRNA biogenesis pathways, namely DROSHA/DGCR8-independent and DICER-independent pathways. In the former, miRNAs are directly exported to the cytoplasm via exportin-1, without Drosha cleavage. In the latter, miRNAs are processed by Drosha from endogenous short hairpin RNA transcripts [60,61]. In both canonical and non-canonical biogenesis pathways, RNA-induced silencing complex (miRISC), consisting of the guide strand and AGO protein, is created [62]. RISC complex is able to identify the complementary sequences within the 3’UTR region of the target mRNA, leading to mRNA instability or repressing their translation [63,64]. MiRNA target recognition occurs through highly conserved heptametrical region located at position 2–8 at the 5′ end of the mRNA, called seed sequence. After recognition, different regulatory mechanisms can occur: mRNA deadenylation, mRNA target cleavage or translational repression [65].

Of note, miRNAs have been identified in different biological fluids, including plasma, serum [66,67], saliva [68], breast milk [69], urine and seminal fluid [70]. Usually, extracellular miRNAs can be enclosed in extracellular vesicles, e.g., apoptotic bodies, microvesicles and exosomes, or associated with proteins—especially AGO2 [71,72,73,74]. Given their stability in several biological fluids, this class of small ncRNAs have been suggested as potential circulating biomarkers of different metabolic diseases, including diabetes [75,76,77].

Due to its biogenesis and structure, a single miRNA is able to bind to several mRNAs which share a 3′ UTR complementarity to the seed sequence; on the other hand, a single mRNA can be targeted and regulated by several miRNAs. Thanks to their regulatory function, miRNAs are involved in a variety of biological, physiological and pathological cellular processes, such as immune response, proliferation, and metabolism. Therefore, they have been linked to the pathogenesis of several diseases, including diabetes [78,79,80,81]. Several studies reported alterations in miRNA expression in several processes involved in the development of type 1 (T1D) and type 2 (T2D) diabetes, including autoimmunity, insulin resistance, insulin secretion and β-cell differentiation [82].

### 2.3. Circular RNAs

CircRNAs are defined as covalently closed RNAs lacking of 3′ polyadenylation [83], highly conserved among species, firstly identified in yeast and in viruses [84,85]. Until a few years ago, circRNAs were considered as useless RNAs, representing by-products of spliceosome-mediated splicing errors (mis-splicing with scrambled exon orders) or intermediates escaped from intron lariat debranching [52]. Usually, pre-mRNA is transcribed by RNA polymerase II (Pol II) and is composed by introns and exons, followed by a 7-methylguanosine cap and poly-adenosine tail, respectively added to its 5′- and 3′-ends. Then, through canonical splicing on 5′-GU and 3′-AG at introns splicing sites, with the assistance of spliceosomes, a pre-mRNA becomes mature and ready to be translated. CircRNAs origin by an alternative splicing mechanism, termed back-splicing. In this process, the 3′-end of an exon binds to the 5′-end of its own or to an upstream exon through a 3′,5′- phosphodiester bond, creating a closed structure with a back-splicing junction site [86,87,88]. Based on the order of splicing events as well as on process intermediates, two models of circRNAs biogenesis were proposed [89] and validated [90]: the lariat model and the direct back-splicing model [88] (Figure 1). Recently, a seminal study extensively described the back-splicing-mediated circRNA biogenesis [91].

Unlike the previously described back-splicing model, lariat-driven circularization occurs following pre-mRNA splicing, when the 3′ hydroxyl of the upstream exon covalently binds the 5′ phosphate of the downstream exon, producing a lariat composed by both exons and introns. The 2′ hydroxyl of the 5′ intron interacts with the 5′ phosphate of the 3′-intron; then, the interaction between the 3′ hydroxyl of the 3′ exon and the 5′ phosphate of the 5′ exon generates an exonic circular RNA (ecircRNA). In general, four main subtypes of circRNAs have been identified: exonic circRNAs (ecircRNAs), mainly derived from single or multiple exons, representing the best identified circular RNA species; circular intronic RNAs (ciRNAs) only containing introns; exonic-intronic circRNAs (EIciRNAs), which consist of both introns and exons; and tRNA intronic circRNAs (tricRNAs), formed by splicing of pre-tRNA intron [92]. As a complex and heterogeneous mechanism, circRNAs biogenesis is tightly regulated at different levels. Among these regulators Intronic Complementary Sequences (ICSs) and RNA Binding Proteins (RBPs), which are respectively cis-elements and trans-factors, should be mentioned [93]. From a functional point of view, circRNAs play several roles. For instance, it has been demonstrated that nuclear circRNAs act as transcriptional regulators at several steps. As an example, some EIciRNAs have been demonstrated to regulate transcription at initiation step [94], while some circRNAs regulate transcription elongation step [95,96]. Alongside transcriptional regulation, cytoplasmic circular RNAs are involved in post-transcriptional regulation, mainly acting as miRNAs sponges. Among circRNAs acting as miRNAs sponges, ciRS-7 is one of the best characterized. Derived from CDR1 (Cerebellar degeneration-related 1) transcript, ciRS-7 shows several binding sites for and regulates miR-7, a miRNA preferentially expressed in endocrine pancreas and, in particular, in β cells [97,98]. As a matter of fact, ciRS-7 inhibition leads to the downregulation of miR-7 target genes, including insulin. Indeed, miR-7 overexpression in MIN6 murine β cell line and in isolated pancreatic islets, induces the upregulation of insulin levels, leading to improved insulin secretion [99].

## 3. Non-Coding RNAs in Metabolic Diseases: State of Art

### 3.1. Long Non-Coding RNAs

#### 3.1.1. Long Non-Coding RNAs and Obesity

Recent bioinformatic techniques, such as RNA sequencing or microarray, enabled the identification of a growing number of lncRNAs involved in obesity and adipocytes differentiation [32,100]. Both gain- and loss-of-function studies strongly suggest a pivotal role of lncRNAs in adipogenesis [36] and a number of lncRNAs have been identified so far, in both human and animal models, as active regulators of different genetic pathways involved in WAT differentiation and functions [101] (Table 1). Historically, SRA (steroid receptor RNA activator) was the first adipogenesis-related lncRNA recognized in murine adipocytes as a co-activator of peroxisome proliferator-activated receptors (PPAR)γ2 [102]. Among lncRNAs involved in adipogenesis, ASMER-1 and ASMER-2, two lncRNAs linked to adipocyte-specific metabolism and associated with obesity and IR, are upregulated in subcutaneous adipose tissue (ScAT). These two lncRNAs are also essential in adipogenesis, lipolysis and adiponectin secretion in differentiated human adipocytes [103]. ADNCR (adipocyte differentiation-associated long noncoding RNA), acting as a competing endogenous RNA for miR-204, leads to overexpression of SIRT-1, which in turn represses adipocyte differentiation and impairs PPARγ pathway in vitro [102,104]. H19 is involved in a regulatory pathway including also CCCTC-binding factor (CTCF), miR-675 and histone deacetylase (HDAC) 4–6, leading to bone marrow mesenchymal stem cells differentiation into adipocytes [102,105] and finally HOTAIR (HOX transcript antisense intergenic RNA) is implicated in pre-adipocytes differentiation [101,106].

Several lncRNAs have been also involved in brown adipose tissue (BAT) regulation. For instance, Blnc1 (Brown fat lncRNA1) was reported by Zhao et al. as being involved in the regulation of thermogenic genes, leading to a higher expression of uncoupling protein 1 (UCP 1) and of mitochondrial genes [3,100,107]. Another example of BAT-associated lncRNA is H19, which was inversely correlated with body mass index (BMI) and positively correlated with browning markers, being also involved in the modulation of adipogenesis, oxidative metabolism and mitochondrial respiration in BAT [101,108]. As BAT activation has been associated with beneficial cardio-metabolic effects, it can be speculated that improving BAT activity or inducing browning of WAT through lncRNA manipulation could represent a promising therapeutic tool for metabolic diseases [3].

Several studies revealed a differential expression of lncRNAs in obese and in non-obese populations. Sun et al., for example, identified a reduced expression of three lncRNAs (lncRNA-p5549, lncRNA-p21015 and lncRNA-p19461), which were inversely correlated with waist circumference, waist to hip ratio (WHR), BMI and fasting plasma insulin levels, in obese but not in lean subjects. In the same study, weight loss induced by a 12-week diet led to the upregulation of lncRNA-p19461 [109].

Indeed, in vitro and in vivo studies indicate that weight loss interventions could affect lncRNA expression profiles. Data on lncRNAs expression in humans following bariatric surgery are scarce, but evidence from animal models [32] suggests changes in lncRNAs’ expression levels, especially those involved in digestion, absorption and inflammatory pathways, after surgical procedures in mice [32,110].

#### 3.1.2. Long Non-Coding RNAs and Type 2 Diabetes Mellitus

In recent years, we are witnessing parallel increases in the prevalence of diabetes mellitus, the spread of obesity and overweight, and increasingly sedentary lifestyles, coupled with an average longer life expectancy [123]. Early diagnosis and effective treatment of diabetes mellitus—one of the top ten causes of death worldwide—have been given increasing attention [7]. In the spectrum of pathogenetic pathways in cardiometabolic diseases, including diabetes, epigenetic phenomena such as DNA methylation, histone modification and altered non-coding RNA expression, are increasingly being studied [9]. Dysregulation of lncRNAs has been demonstrated in animals models and in human pancreatic islets [3,124]. In addition, several lncRNAs participate in different steps of insulin secretion and are associated with development of IR [124] (Table 1). Furthermore, their encoding genes are located near islet-specific chromatin domains containing genes involved in modulation of β-cell function or mapped to diabetes susceptible loci [3,125]; however, their specific function and mechanisms of action are still not known in details [124].

In subjects with T2D, MetS and low HDL levels, a reduced expression of metastasis-associated lung adenocarcinoma transcript 1 (MALAT1) was described in serum and in circulating exosomes, together with overexpression of H19 in patients with poor glycaemic control compared to subjects with glycated hemoglobin <7%. Moreover, MALAT1 has been implicated also in angiogenesis in diabetic eye and kidney [112]. In a study comparing T2D patients with healthy controls, authors found dysregulated levels of some lncRNAs in peripheral blood. Interestingly, these dysregulated lncRNAs were positively correlated with IR, impaired glucose control, inflammation and transcriptional markers of senescence, and significantly associated with T2D, even after adjusting for confounding factors [9]. Similar data were found in newly diagnosed T2D patients, suggesting that aberrantly expressed lncRNAs regulate IR and inflammation, leading to impaired glucose homeostasis [126].

LncRNAs have also been investigated in micro- and macrovascular diabetic complications. Among lncRNAs related to diabetic complications, one of the most -studied is ANRIL (antisense noncoding RNA in the INK4 Locus or CDKN2B-AS1), regarded as a putative biomarker of cardiovascular risk and atherosclerosis in diabetes. ANRIL genetic polymorphism SNP rs10757278 has been associated with risk of major adverse cardiovascular events [116]. Moreover ANRIL is implicated in myocardial apoptosis and fibrosis leading to progression and evolution of acute myocardial infarction in T2D patient [117]. As additional examples, MALAT1 has been involved in increased ROS and pro-inflammatory cytokines expression, leading to endothelial damage, both at micro- and macrovascular level [123], while MEG3 (maternally expressed 3 lncRNA) is reduced in retina during hyperglycemia [116]. With regard to diabetic nephropathy, MALAT1 and TUG1 (Taurine-upregulated gene 1) in animal models and LINC01619 in human renal biopsies, appear to be dysregulated in diabetic podocytopathy [124]. In a study involving diabetic patients with chronic complications, compared to healthy controls, CASC2 (Cancer Susceptibility Candidate 2) was downregulated in serum and in renal tissue of T2D patients with chronic kidney disease (CKD); additionally, T2D patients without complications but with low CASC2 levels, had higher incidence of renal failure in the following 5 years, suggesting its potential use as a diagnostic biomarker in CKD [118]. Furthermore, up-regulation of MIAT and MALAT1 was found in kidney samples of both diabetic patients and animal models [124]. The role of lncRNAs has been also explored in diabetic peripheral neuropathy. In particular, Yu et al. demonstrated that NONRATT021972 was up-regulated in T2D subjects with worse symptoms related to neuropathic pain, together with increased TNF-α levels. This observation was further supported by evidence showing that small interference LncRNA (siRNA) NONRATT021972 lowered blood glucose and mitigated inflammation by decreasing TNF-α in rats, thus relieving neuropathic pain. These data pave the way for potential treatment for neuropathic pain [119]. Finally, lncRNAs appear to modulate the onset of diabetic gastroparesis; specifically, MALAT1 was found to be over-expressed in animal models of gastroparesis and in T2D patients suffering from gastroparesis-related symptoms, and its effect may be linked to smooth muscle cells [113].

#### 3.1.3. Long Non-Coding RNAs and NAFLD/NASH

As already reported, NAFLD prevalence is continuously growing, despite several cases remaining undiagnosed because routine screening for NAFLD is not yet recommended. It has been reported that NAFLD is associated to lncRNAs, whose functions and roles are not always clear, as for other liver diseases [121,127]. Several authors described altered circulating and hepatic expression pattern of different lncRNAs in subjects with hepatic IR and steatosis [37] (Table 1). In 1998, H19 was the first lncRNA described as involved in liver disease [127] and twenty years later Liu et al. discovered that H19, together with PTBP1 (Polypyrimidine tract-binding protein 1), was upregulated by fatty acids (FAs) in hepatocytes and in diet-induced fatty liver. High fat diet (HFD) favors lipogenesis through SREBP1c (sterol regulatory element-binding protein 1c), but this effect breaks off when H19 or PTBP1 are deleted [111]. Another lncRNA involved in hepatic steatosis is lncARSR, which is upregulated in patients with NAFLD and has been suggested as potential therapeutic target, given that its knockdown improved hepatic lipid accumulation, both in vivo and in vitro [120,128]. LncRNAs may also be involved in the progression to NASH and fibrosis. Gene expression profiling of nearly 5000 lncRNAs performed in liver samples of obese patients with steatosis or NASH, and healthy subjects, highlighted that lnc18q22.2, a liver-specific lncRNA (RP11-484N16.1), was associated with NASH severity, lobular inflammation and NAFLD activity score and that a decreased cell survival was observed upon lnc18q22.2 silencing, suggesting an anti-apoptotic effect of this lncRNA in hepatocyte [121,122]. Leti et al. demonstrated the over-expression of three lncRNA (i.e., nuclear paraspeckle assembly transcript 1, NEAT1; hepatocellular carcinoma upregulated lncRNA, HULC; MALAT1) in patients with advanced liver fibrosis compared to NAFLD patients with steatosis and/or lobular inflammation [114]. Furthermore, MALAT1 was demonstrated to target C-X-C motif chemokine ligand 5 (CXCL5), whose transcript and protein levels were increased in fibrotic liver and activated hepatic stellate cells, supporting the hypothesis that MALAT-1 has a pivotal role in the development of steatohepatitis and fibrosis in NAFLD patients [114,115]. Other authors focused on MEG3’s increased hepatic levels in NASH and fibrosis in NAFLD patients, and on APTR’s higher expression in fibrotic liver, both in humans and animal models, and in serum of cirrhotic patients [21]. It is also worth mentioning that variants in lncRNAs influence NAFLD susceptibility and severity, as in the case of the rs2829145 A/G located in lnc-JAM2-6, linked to a worse metabolic profile [129].

### 3.2. Micro RNAs

#### 3.2.1. MicroRNAs and Obesity

An impaired expression of different miRNAs could play a pivotal role in the pathogenesis of metabolic diseases. Several studies have demonstrated the presence of multiple loci associated with obesity and MetS in human genome. Kunej et al. demonstrated that 221 out of 1736 obesity-associated loci coincided to microRNAs [130].

MiRNAs have been shown to modulate pathways controlling adipogenesis [131,132], inhibiting or accelerating adipocyte differentiation, which has been reported to be impaired in obesity (Table 2). Thus, miRNAs dysregulation could participate to metabolic processes underlying obesity development [131]. Indeed, the adipose tissue-derived vesicles enriched of miR-27a, miR-34a, miR-141-3p, miR-155, miR-210 and miR-222 are involved in the development of IR during obesity [133]. Most of the available data on miRNAs expression during adipogenesis are derived from studies in cellular and animal models, but several authors have evaluated miRNA expression in human adipose depots and correlated miRNAs levels with key metabolic parameters, such as BMI, glycemia, or leptinemia. A differential expression of miR17-5p, miR132 and miR134 was demonstrated in omental fat from overweight/obese T2D patients compared to normoglycemic patients. In particular, the expression of miR-17-5p and miR-132 was negatively associated with visceral fat area [131]. Furthermore, other authors observed a positive correlation of miR-17-5p and miR-132 expression and glycosylated hemoglobin, leptin, BMI and fasting blood glucose, in omental fat and blood from obese patients compared to non-obese individuals [134]. Ortega et al., through miRNA expression microarray global analysis, showed that miR130b, miR210, miR221, miR125b and miR100 were down-regulated during adipocyte differentiation; while miR130b and miR210 were also down-regulated in ScAT depots of obese patients, the others were highly expressed in patients with obesity [132]. In 2011, Lee and colleagues confirmed a downregulation of miR-130 expression in the abdominal ScAT of obese women compared to lean women. Authors showed also a correlation between miR-130 downregulation and the increase of PPARγ mRNA levels, a major regulator of adipogenesis, suggesting that this miRNA reduced adipogenesis through the repression of PPARγ and that this deregulation was linked to human obesity [135]. In contrast, another study demonstrated that miR-130b expression was overexpressed in plasma of obese children, and directly related to BMI and other indicators of obesity, suggesting that some miRNAs could be deregulated in prepubertal obesity [136].

Several studies also investigated differences in extracellular miRNAs expression patterns between normal and obese condition. In ScAT obtained from subjects with different degrees of obesity, expression of miR-21 was more elevated in subjects with a BMI > 30 and positively correlated with BMI, while in patients with a BMI > 30 the expression of miR-143 was reduced compared to lean subjects [157]. In another study investigating the expression of circulating miRNAs in adult males with varying degrees of obesity, authors observed a substantial increase of miR-140-5p, miR-142-3p and reduced levels of miR-15a, miR-423-5p, and miR-520c-3p in morbidly obese patients. These results suggest a potential role of these miRNAs as novel biomarkers for risk assessment and classification of subjects with morbid obesity [158]. Pescador et al., demonstrated that in obese patients serum concentrations of miR-503, miR-376a and miR-138 were reduced and miR-15b levels was elevated compared to that observed in control subjects [159]. In another study, comparing lean and obese individuals, the latter showed markedly higher circulating levels of miR-122, whose expression increased as the degree of obesity worsened. Moreover, circulating miR-122 positively correlated with gender, BMI, WHR, fat percentage, blood pressure, liver enzymes, TG, HDL-c, fasting and 2 h post load blood glucose and insulin, and HOMA-IR [160]. In a pediatric cohort, a deregulation of several miRNAs (i.e., overexpression of miR-31-5p and miR-2355, and a downregulation of miR-206) involved in lipid metabolism and adipocytes differentiation, was identified in overweight or obese children compared to normal weight pairs [161]. Recently, Brovkina et al., investigated transcriptional differences between obese subjects with and without T2D by RNAseq and qPCR analysis, showing an overexpression of five miRNAs (miR-204-5p, miR125b-5p, miR-125a-5p, miR320a, miR-99b) in patients with obesity and T2D, while miR-23b-3p and miR197-3p were upregulated in obese subjects without diabetes [162]. Evidence of differential miRNA expression between obese and lean subjects prompted researchers to explore the potential of miRNAs for risk prediction of obesity and obesity-related complication. Cui et al., for example, through miRNA high-throughput sequencing, investigated microRNAs that could be predictive of T2D in adulthood in obese children, demonstrating that three circulating miRNAs (miR-486, miR-146b and miR-15b) resulted upregulated in both obese children and adult T2D patients. In particular, miR-486 was found to accelerate preadipocyte proliferation and myotube glucose intolerance, while miR-15b and miR-146b seemed to suppress glucose-stimulated insulin secretion (GSIS) [137].

Interestingly, recent findings indicate that weight loss can modulate the expression of some miRNAs. Many studies reported an association between diet and lifestyle with microRNAs expression, and several miRNA families (i.e., miR-21/ 590-5p family, miR-17/20/93 family, miR-221/222 family, miR-200b/c family, let-7/miR-98 family and miR-203) were deregulated following dietary manipulation [163]. Recently, J.W. Helge and colleagues, measured the expression of miRNAs in ScAT from severely obese patients by genome-wide microarray analysis, before and after a weight loss intervention, demonstrating an upregulation of miR-29a-3p and miR-29a-5p and a downregulation of miR-20b-5p expression, 15 weeks after treatment [164]. In another study, acute weight loss reversed the expression pattern of dysregulated circulating miRNAs in obese women, moving to levels observed in lean individuals [165]. These data suggested that circulating miRNAs could be useful biomarkers of weight loss effect in obesity.

#### 3.2.2. MicroRNAs and Type 2 Diabetes Mellitus

Multiple studies have highlighted a pivotal role of miRNAs in the molecular mechanisms contributing to loss of functional β-cell mass, such as β-cell dysfunction (i.e., impaired insulin production and secretion), differentiation, survival and apoptosis, which ultimately lead to T2D [166] (Table 2).

Among islet-specific miRNAs, mir-375 is the best characterized and evolutionarily conserved. Poy et al. demonstrated that miR-375 overexpression repressed GSIS, reducing insulin gene transcription; conversely, downregulation of this miRNA led to increased insulin secretion. Consistent with this study, miR-375 expression in pancreas of T2D donors was higher than in healthy individuals, highlighting its fundamental role in T2D pathogenesis [138]. In addition, available evidences showed a deregulation of miR-375, together with other miRNAs, up to 5 years before the onset of T2D and pre-diabetes, suggesting a role for disease prediction and prevention in individuals at high risk of T2D [167].

Mir-200 miRNA family, consisting of five miRNAs (miR-200c, miR-141, miR-200a, miR-429 and miR-200b), has been implicated in the protection against β-cell apoptosis in vitro and in β-cell dedifferentiation [168]. The deletion of miR-200c (among the most expressed miRNAs in β-cells) partially protected from β-cell death following oxidative stress, while ablation of all miRNAs belonging to this family resulted into greater protection from β-cell apoptosis. Therefore, miR-200 family is critical for diabetes pathophysiology [169].

MiR-7 represents another miRNA enriched in mature human pancreatic islets which negatively regulates GSIS, by inhibiting expression of genes involved in insulin granules fusion with plasma membrane and SNARE proteins [97]. It has been demonstrated that hsa-miR-7–1-3p levels were downregulated in pancreatic islets of T2D vs non-diabetic donors, while hsa-miR-7-3-5p expression levels were upregulated [139].

Among miRNAs specifically expressed in pancreatic islets and particularly in β-cells, miR-184 is mainly involved in the regulation of insulin secretion and compensatory proliferation during IR [139,140]. MiR-184 knockout in β-cells, leads to increased β-cell proliferation, thus improving insulin secretion after glucose challenge. In rat and in human islets, blockage of this miRNA protects β-cell against apoptosis induced by prolonged exposure to FAs and/or proinflammatory cytokines [140]. Interestingly, miR-184 expression was negatively correlated to insulin secretion and positively to insulin mRNA levels in human islet and was reduced in pancreatic islets of T2D donors [140]. Recently, Grieco et al. showed that NKX6.1, a transcription factor altered in pancreatic islets of T2D donors, directly regulates expression of miR-184. Moreover, NKX6.1 downregulation as well as its nucleus-to cytoplasm translocation leads to a reduced expression of miR-184, with consequent overexpression of its target gene CRTC1, thus leading to β-cell protection from apoptosis [170].

Other miRNAs have been implicated in de-differentiation and protection against apoptosis of β-cells, such as miR-24—which has been attributed a protective role against apoptosis during metabolic stress [141].

MiR-204 represents another miRNA involved in the regulation of insulin secretion. Guanlan Xu et al. identified a novel TXNIP/miR-204/MafA/insulin pathway in which the upregulation of thioredoxin-interacting protein (TXNIP), a cellular redox regulator, induced the expression of miR-204, which in turn inhibits insulin production by negatively regulating MafA, a transcription factor essential for insulin transcription [142]. MiR-204 also targets and downregulates glucagon like peptide 1 receptor (GLP1R) and its deletion enhanced islets GLP1R expression and function, resulting in improved glucose tolerance and insulin secretion [143] (Figure 2).

Finally, some researchers hypothesized a role for miRNAs as biomarkers for predicting susceptibility to diabetes. As an example, the dysregulation of miR-124a, which is involved in β-cells differentiation, seems to highly predispose to T2D [171].

#### 3.2.3. MicroRNAs and NAFLD/NASH

Increasing evidence has been provided that some miRNAs regulate hepatic molecular pathways, such as lipid metabolism, inflammation and oxidative stress, thus playing a pathophysiological role in NAFLD/NASH development [172,173]. Dysregulation of several miRNAs can lead to the perturbation of several steps of hepatic fat metabolism, including general mechanisms of biosynthesis and catabolism of lipids and cholesterol regulation, thus converging to aberrant accumulation of lipids in hepatocytes [172].

MiRNAs with established regulatory functions in hepatic metabolism include: miR-122, miR-21, miR-29a, miR-33 and miR-34a [174]. Of note, additional deregulated hepatic or circulating miRNAs in NAFLD/NASH can be included in a non-exhaustive list, such as: miR-192, miR-375, miR-146b, miR-221/222, miR-132, miR-181, miR-422, miR-139 and miR-197 [172] (Table 2).

MiR-122 is the most enriched miRNA in the liver, representing almost 70% of all miRNAs copies expressed in this tissue [175]. MiR-122 contribute to hepatocyte maturation and proliferation, through stimulation of liver specific genes, including the Hepatocyte Nuclear Factor 6 (HNF6) [144]. MiR-122 is highly expressed in healthy liver, but conversely, is significantly reduced in damaged or unhealthy hepatic tissue [176]. As a matter of fact, decreased hepatic levels of miR-122 has been observed in NASH patients compared to healthy controls [177], while miR-122 serum levels are reported to be increased in NASH/NAFLD [175,178]. These opposing changes could be explained by a FAs-dependent mechanism. In fact, Chai and colleagues, using different mice models, speculated that free fatty acids (FFAs), through RORα pathway, induced hepatic miR-122 expression and its subsequent secretion, thus explaining its increased secretion in the blood. They observed that circulating miR-122 can reduce triglyceride synthesis in extra-hepatic tissues (i.e., skeletal muscle and adipose tissues), producing a crosstalk between the liver and distant tissues [145]. The appearance of a NAFLD phenotype in miR122 KO mice corroborate the “anti-NASH functions” of miR-122. This NAFLD phenotype is partly a consequence of miR-122 target genes upregulation, but it is also the result of alterations in lipid secretion, increased lipogenesis, tumor necrosis factor alpha (TNF-α), elevation of chemokine (C-C motif) ligand 2 (CCL2), interleukin 6 (IL-6) and macrophage recruitment [179,180]. Subsequent studies also observed that miR-122 inhibition by antagomiR-122 exacerbated fatty liver in high-fat diet (HFD)-fed mice by decreasing β-oxidation [145].

Similarly to miR-122, miR-29a has a significant role in regulation of genes implicated in many liver diseases, especially liver malignancy, fatty liver disease and MetS [181]. MiR29a hold significant diagnostic relevance in NAFLD [182]: Jampoka and colleagues observed a significant downregulation of miR-29a in serum from patients with NAFLD compared to healthy subjects, making it a highly sensitive and specific diagnostic biomarker for NAFLD. The role of miR29a in the pathogenesis of NAFLD is also supported by the finding of miR29a-mediated repression of lipoprotein lipase (Lpl)—a functional enzyme involved in lipids uptake from the bloodstream—in hepatocytes. Thus decreased hepatic miR-29a levels could induce an increase of intracellular lipids accumulating in liver [146].

Unlike the above mentioned miRNAs, high levels of miR21 have been described both in liver and plasma of NASH patients [177,183], but is inactivated in physiological condition [184]. Recent studies revealed a key role of miR-21 in inflammation and hepatic metabolism. In NAFLD, miR-21 seems to regulate triglycerides, free cholesterol, and total cholesterol levels, through the inhibition of 3-hydroxy-3-methyl-glutaryl-coenzyme A reductase (HMGCR) [147], and fatty acid-binding protein7 (FABP7), which induces FAs uptake and accumulation [149]. MiR-21 targets include factors involved in suppressing the development of liver steatosis, particularly phosphatase and tensin homolog (PTEN), which inhibits DNL and FAs uptake [150] or PPARα, triggering lipid oxidation [148]. As a confirmation of its role in lipid metabolism, it was recently demonstrated that specific ablation of miR-21 in hepatocytes can suppress steatosis development in HFD mice, through the upregulation of several miR-21 targets involved in lipid metabolism [172,184].

Another miRNA which has been demonstrated to be upregulated in liver and blood of NAFLD and especially in NASH patients, is miR-33 [185]. As a matter of fact, this miRNA has been suggested as a therapeutic target to manage both NAFLD and Mets, given that it is deeply involved in both cholesterol and FAs metabolism, by targeting key enzymes in cholesterol synthesis pathway (i.e., ABCA1 and ABCG1, CPT1A and AMPKα), and in glucose metabolism, by inhibiting gluconeogenesis through the modulation of phosphoenolpyruvate carboxykinase (PCK1) and glucose-6-phosphatase (G6PC) [151,152,153,154,155].

Similarly, miR-34a has been also shown to be upregulated in liver and serum of patients with NAFLD [177,186,187,188], making it a valid and reliable biomarker able to distinguish patients with NASH from those with NAFLD [178,189]. In vitro and in vivo studies in mice observed that miR-34a specifically targets PPARα and Sirtuin 1 (SIRT1), thereby suppressing FAs catabolism and eliciting steatosis. In addition, miR-34a inhibition seems to enhance AMP-activated protein kinase α (AMPKα) function, one of the main antagonist of lipogenic pathway [156].

Overall, these data suggest that some dysregulated miRNAs can promote liver disease onset and progression, while others can act in opposite way, improving defensive responses.

### 3.3. Circular RNAs

#### 3.3.1. Circular RNAs and Obesity

The first study identifying a potential role in essential molecular mechanisms in adipose tissue was published by Li and colleagues, who demonstrated that circRNA_1897 and circRNA_26852 were highly downregulated in subcutaneous tissue of large White pigs and Laiwu pigs. In the same study, authors revealed that circRNA_1897 directly targets miR-27a and miR-27b-3p, while miR-874 and miR-486 are bound and targeted by circRNA_26852 [190]. Importantly, miR-874 and miR-486 are strongly involved in pathways associated with adipocytes differentiation and lipid metabolism [191]. On the other hand, miR-27a and miR-27b-3p are mainly involved in lipolysis and in inhibition of adipocyte differentiation via a PPARγ-dependent mechanism [192]. Alongside studies performed in several animal models, in the last three years, different works were published about the role of circRNAs in human adipogenesis, lipid metabolism and related disorders (Table 3). For instance, Guo and colleagues identified a strong downregulation of circ_0046367 in HepG2, an hepatoma human cell line, treated with high concentrations of oleate and palmitate [193]. As a matter of fact, Guo et al. reported that circ_0046367 abolishes the inhibitory effect of miR-34a on PPARα, thus leading to the translocation of this protein from cytoplasm to nucleus, with the consequent activation of genes involved in lipid metabolism such as Cpt2 and Acbd3 [193]. Another circRNA mainly tested in oleate-stressed HepG2 cells is circHIPK3. In details, this circRNA enhances the lipid droplets accumulation following oleate treatment, mainly acting as miRNA sponge for miR-192-5p and consequently targeting on Foxo1 [194]. The regulation of miRNAs by circRNAs has also been demonstrated for circCDR1as on miR-7-5p. Indeed, it has been reported that circCDR1as/miR-7-5p/Wnt5 axis is able to enhance adipogenic differentiation while impairing osteogenic differentiation of Bone Marrow-derived Stem Cells (BMSCs) [195]. Another potentially important circRNA in obesity is circH19, which resulted upregulated in serum of patients with MetS and associated to obesity-related clinical parameters (positively correlated to BMI and negatively correlated to HDL concentration). Furthermore, the inhibition of circH19 leads to an enhancement of lipids droplets formation and adipogenesis in Adipose Derived Stem Cells (ACDCs), as well as to an increased expression of adipogenic markers such as Pparγ, Srebp1c and Cebp1-α. Finally, a recent study by Arcinas and colleagues identified circ_Tzhz2-1 and circ_Arhgap5-2 as highly upregulated during differentiation of primary white adipocytes. Importantly, the inhibition of circ_Arhgap5-2 resulted in lipid accumulation and reduction of adipogenesis markers such as Pparγ, Cebpα, Fabp4 and AdipoQ [196].

#### 3.3.2. Circular RNAs and Type 2 Diabetes Mellitus

Most circRNAs related to T2D have been mainly identified as potential biomarkers for disease prediction as well as involved in T2D pathogenetic mechanisms (Table 3). Among these, the first study identifying an altered expression of several circulating circRNAs in both pre-diabetics and T2D patients was published by Zhao and colleagues [198]. In this study, authors evaluated circRNAs expression in whole blood of non-diabetic (CTR), pre-diabetic and T2D patients. Microarray analysis revealed the downregulation of 411 circRNAs and upregulation of 78 circRNAs, which were further considered as potential biomarkers. Among these latter, hsa_circ_0054633 resulted the most promising, as its expression resulted gradually increased from CTR to T2D group; moreover, ROC analysis revealed high specificity and sensitivity values, independently on sex and age. Therefore, hsa_circ_0054633 could represent a promising biomarker for the prediction of T2D onset in the early stage of pre-diabetes. In another study, Fang and colleagues [199] reported the results of a RNAseq experiment performed on RNA extracted from blood of 5 T2D patients and 5 healthy subjects. Following sequencing data analysis, authors detected a differential expression on 220 circRNAs, among which 107 upregulated and 113 downregulated. Validation of differentially expressed circRNAs confirmed the upregulation of circANKRD36, further correlated with glucose and glycated haemoglobin levels of patients. Importantly, circANKRD36 was not expressed in plasma, but enriched into blood cells and positively correlated with plasmatic IL-6 levels [199], thus indicating that this circRNA could potentially play a role in inflammatory mechanisms occurring in T2D.

A more recent study published by Stoll et al. [197] extensively investigated the role of a new circular RNA, essential for β-cell function, insulin production and secretion. In details, using a two-algorithm computational approach, authors identified several circRNAs generated from linear transcripts of important β-cell genes such as Pcsk2 (Proprotein Convertase Subtilisin/Kexin Type 2), Gck (Glucokinase) and most importantly Insulin (human INS, murine Ins2). The lariat deriving from human INS and mouse Ins2, called ci-INS and ci-Ins2 respectively, were only detected in β-cells. Most importantly, ci-INS knock down in cultured human islets is able to reduce insulin secretion following glucose and KCl stimulation, mainly through the regulation of several genes involved in insulin secretion such as SYT7 (Synaptotagmin-7), PCLO (Piccolo Presynaptic Cytomatrix Protein), CACNA1D (Calcium Voltage-Gated Channel Subunit Alpha1 D) and UNC13A (Unc-13 Homolog A). As a matter of fact, authors finally demonstrated that ci-INS is strongly downregulated in human islets from T2D donors and negatively correlated with HbA1c levels.

#### 3.3.3. Circular RNAs and NAFLD/NASH

The pioneer research group investigating the role of circRNAs in liver disease is that of Guo and colleagues. Indeed, they firstly performed a microarray profiling on HepG2 cells stressed with palmitate and oleate in order to reproduce typical conditions of fatty liver disease, identifying the differential expression of 357 circRNAs mainly involved in pathways related to steatosis. Among these, hsa_circRNA_021412 resulted the most interesting, as its downregulation leads to the upregulation of miR-1972, consequently inhibiting Lipin1 (LPN1) and resulting in downregulation of long chain acyl-CoA synthetases and in development of hepatic steatosis [200] (Table 3). The same authors, a few months later, showed a reduced expression of hsa_circRNA_0046367 following FFA-induced steatosis in HepG2 human cell line. Further investigation revealed that hsa_circRNA_0046367 acts as miRNA sponge on miR-34a [193], a miRNA largely studied as potential biomarker for liver diseases [186,188], consequently abolishing its inhibitory effect on PPARα and leading to steatosis. On the other hand, restoration of hsa_circRNA_0046367 resulted in a prevention of steatosis onset due to PPARα inhibition by miR-34 [193]. Interestingly, miR-34a/PPARα pathway has also been demonstrated to be targeted by another circRNA, namely hsa_circRNA_0046366, also reduced in steatotic HepG2 human cells. Most importantly, authors also demonstrated that PPARα restoration is able to promote transcriptional activation of several genes involved in lipid metabolism, such as CPT1A and SLC27A, thus leading to steatosis improvement [201] (Table 3).

## 4. Potential Clinical Application of Non-Coding RNAs

The lack of shared and reliable tools to assess IR limits the possibility of an early diagnosis and identification of high-risk individuals, before developing metabolic alterations. Therefore, a number of subjects remain undiagnosed [37]. In recent years ncRNAs has been increasingly studied in metabolic disorders [13,202]. As discussed above, aberrant expression of several ncRNAs and their involvement in IR and metabolic diseases has been proved in several studies. Therefore, their clinical application is highly envisaged. Indeed, ncRNAs are extremely attractive candidates as diagnostic and predictive biomarkers, given their resistance to degradation, stability, and easy detection in biological fluids [21]. Further data are undoubtedly needed to strengthen available observations and evaluate the effective applicability in clinical settings. Additionally, given the complexity of IR pathogenesis, an approach based on a combination of multiple biomarkers should be preferred to ensure higher diagnostic accuracy [5].

Among ncRNAs, miRNAs are the most extensively studied as disease biomarkers. It has been repeatedly reported that metabolically impaired and normal weight subjects display distinct profiles of circulating miRNAs. MiRNA expression profiling has been often performed in order to find new biomarkers for metabolic diseases. However, use of different profiling platforms and different operative procedures, as well as differences in study population and tissues analyzed, led to inconsistent results and nonreproducible data [203]. Therefore, standardized samples collection protocols and consistent analytical procedures are strongly needed [204]. Several miRNAs signatures have been proposed as diagnostic tools for obesity, diabetes and their metabolic complications. For instance, Ortega et al. provided evidence of a specific circulating miRNA signature in morbidly obese men, strongly linked to adiposity markers, which changed along with significant surgery-induced weight loss [158]. Interestingly, some miRNAs already identified in studies among adults were also confirmed to be dysregulated in more than one report of obese children and adolescents, with or without metabolic impairment [205]. The first evidence of a potential use of miRNAs in the diagnosis and follow-up of T2D was provided by Zampetaki and colleagues, who reported altered expression of several miRNAs in T2D patients compared with controls, and showed that the altered expression was detectable years before disease onset, thus representing an interesting tool for disease prediction, especially in high-risk populations [206]. In a recent meta-analysis, Zhu and Leung identified 40 significantly and consistently dysregulated miRNAs, out of more than 300 differentially expressed miRNA reported in 38 studies comparing humans and/or animals with and without diabetes, and suggested a set of ten miRNAs as disease biomarker for T2D—including circulating (miR-103, miR-107, miR-132, miR-144, miR-142-3p, miR-29a, miR-34a and miR-375) and tissue (miR-199a-3p and miR-223) biomarkers [203]. Similarly, Seyahn and colleagues showed that subjects with prediabetes were best distinguished from healthy controls by assessing circulating miR-146a, miR-126, miR-30d, and miR-148a, while T2D subjects were best distinguished by measuring miR-30d and miR-34a levels [207]. A potential practical application may be recognized to some circulating miRNA, which are correlated with HbA1c (miR-499, miR-103, miR-28, miR-29a, miR-9, miR-30a-5p, miR-150), or are associated with hyperglycemia and IR (miR-802) and diabetic vascular complications (miR-9, miR-370, miR-143, miR-145), thus may be used to predict T2D onset [21,37,100,116]. The miRNA signature has also been extensively studied in NAFLD, as circulating endogenous miRNAs could represent attractive non-invasive biomarkers to accurately diagnose and monitor NAFLD and liver fibrosis [208]. Different miRNA panels for NAFLD diagnosis have been proposed [23,175,209] and, once validated, they could hopefully replace liver biopsy, which is still the mainstay of diagnosis and monitoring of NAFLD, though limited by its invasiveness, expensiveness and risk of complications [208,210]. Almost all the diagnostic panels proposed for NAFLD include miR-122, as the most widely studied liver-specific miRNA, with 75% sensitivity and specificity above 80% [21,182,211]. Specific miRNAs signature might also be used as risk estimation of better or worse prognosis, and to predict efficacy in therapeutic interventions. Indeed, it has been shown that weight loss interventions, particularly bariatric surgery, and antidiabetic drugs could modify miRNA expression profiles [32,100,158].

Assessment of lncRNAs as biomarkers for pre-diabetes and T2D is ongoing. For example, growth arrest specific 5 (GAS5) has been suggested as a prognostic biomarker, as reduced GAS5 levels increase the risk of developing diabetes. Similarly, ENST00000550337.1 levels may be able to differentiate between pre-diabetes and T2D, and circulating H19 levels seem to discriminate patients with better glycemic control from those with poorly controlled diabetes [37,212]. A number of lncRNAs have also been involved in liver disease and have been associated with NAFLD development and progression [21].

Current knowledge on the role of circRNAs in IR-related diseases is relatively limited. However, preliminary data on the contribution of circRNAs in pathophysiology of diabetes and its related cardiovascular complication, have encouraged to explore circRNA profiles, both in tissue and blood, as valid biomarker for the diagnosis and prognosis of diabetes [202]. For instance, several authors demonstrated the potential use of circ-0054633 as low-cost, specific and sensitive diagnostic biomarker for prediabetes and T2D, as circulating levels of this circRNAs gradually increased from normoglycemia to pre-diabetes up to T2D [198,213]. Other circRNAs have been suggested as predictive biomarker of micro- and macrovascular complications in diabetic patients. As for NAFLD, recent findings suggest that aberrant signaling of circRNA_0046367 and circRNA_0046366/miR-34a/PPAR may be involved in steatosis and could represent a therapeutic target in NAFLD therapy [21]. Other studies identify other circRNAs associated with hepatic steatosis, such as circScd1—significantly lower in NAFLD—and the circRNA_021412/miR-1972/LPIN1 signaling pathway, involved in liver metabolism and, potentially, in steatosis [13,21,214].

Apart from their potential application as diagnostic biomarkers, ncRNAs have also been investigated as therapeutic targets [215]. Indeed, the relevance of ncRNAs as transcription factors makes them suitable as therapeutic agents, exploiting their gene silencing potential [32].

Actually, miRNA agonists and antagonists could represent interesting therapeutic tools, to restore altered miRNA expression in specific tissues [100]. In case of downregulation of miRNA, a therapeutic strategy would be represented by transfection of synthetic miRNA mimetics (miRNA mimics) or plasmid/viral vectors, to achieve a pharmacological activation of miRNA function, whereas in case of miRNAs overexpression, the therapeutic strategy would be to transfect specific synthetic antisense miRNA/oligonucleotides (anti-miRs or antagomiRs), inducing a decrease in miRNA levels by inhibiting intracellular endogenous miRNAs [100]. CircRNAs can be used also as miRNAs inhibitors, through the elimination of multiple miRNAs. Furthermore, altered expression of lncRNAs can be silenced or restored for therapeutic purposes [100]. For example, lncRNA levels can be inhibited through short interfering RNAs (siRNAs), specifically binding to complementary sequences and inhibiting expression of lncRNA targets, or antisense oligonucleotides (ASO), blocking lncRNA activity. Both siRNAs and ASO can be also used to disrupt secondary structure of lncRNAs [212]. Lastly, combining miRNA or lncRNA-targeting therapeutics may represent a more effective option to boost therapeutic efficacy, as multiple molecular pathways underlie the development of metabolic syndrome [216].

The translation of preclinical results into clinical trials, demonstrating feasibility and safety of ncRNA-based therapies, is still underway. Indeed, some encouraging preclinical data regarding miRNA-based therapy are derived from animal models of IR. NcRNA-based therapy in metabolic diseases raised great expectations. In the context of diabetes therapy, for example, some authors suggested to use ncRNAs with a key role in β-cell function to prevent β-cell failure and apoptosis [13,202,214]. Indeed, ciRS-7 overexpression seems to improve insulin secretion [13,202,214]. Other authors also suggest ncRNA-based therapy to prevent diabetic microvascular complications, for instance by addressing angiogenesis and endothelial proliferation. Indeed, MEG3 upregulation seems to lessen retinal angiogenesis [116,214] while circ_0005015 silencing attenuates endothelial proliferation in human retina [10,13,214]. In murine models, the inhibition of miR-143/145 significantly reduces the progression of atherosclerotic plaque [116]. Another promising therapeutic strategy might be represented by the inhibition of fibrosis, both by antagonizing ncRNAs with profibrotic effect (e.g., miR351 [216], miR141, circ000203 [10]) or through overexpression of antifibrotic ncRNAs (e.g., miR29 [216]). The manipulation of ncRNA expression has been also suggested in NAFLD [23]. Just to give some examples, miR-122 inhibition reduces plasma cholesterol levels and might represent a therapeutic approach in early stages of NAFDL, whereas miR-34a inhibition prevents lipid accumulation in liver and might be used in NASH patients given its role in regulating oxidative stress and inflammation and its inhibition seemed to prevent lipid accumulation [21,23]. Further, the inhibition of miR-499 has been related to improvement in NAFLD [37]. Finally, miR21 inhibition has proven beneficial in obesity and metabolic syndrome [217].

Although no specific ncRNA-based therapy to treat MetS is currently available [100,217], pharmaceutical companies have become interested in the field and some interesting molecules are in the pipeline. For example, antagoMir-103/107 is being evaluated for T2D with NAFLD in a phase I/IIA clinical trial (NCT 02826525) [216].

However, clinical translation into diagnostics is delayed by several limitations. First of all, available correlational and merely descriptive data alone are not sufficient proof of a causal relationship [21,202]. Another limit is represented by discrepancies between studies, potentially leading to incorrect conclusions. As a matter of fact, the need for studies with larger sample sizes and greater homogeneity has been emphasized, to lessen the high variability in the profiles of ncRNAs described, in order to draw firm conclusions [202,205]. Low concentrations might also hinder an accurate quantification of circulating ncRNAs—especially miRNAs—despite the availability of sensitive detection methods [21]. Moreover, a large number of data on ncRNAs expression profiles in metabolic diseases are derived from animal studies, but whether data can be applicable in humans has not yet been conclusively proven. The main challenges to be addressed before the development of ncRNAs-based therapy include effective delivery in recipient tissues, and the considerable risk of side effects and off-target effects, as a single miRNA might have multiple targets within a specific tissue and affect other miRNAs’ expression [100,216,217].

## 5. Conclusions

The prevalence of IR-related metabolic diseases has dramatically increased in recent years, reaching epidemic proportions, and it is expected to increase further, representing a public health issue worldwide. Diagnosis and treatment are often challenging and require prompt solutions to manage this pandemic. Major efforts to better define the molecular pathway leading to IR are essential, as it represents the underlying pathogenic mechanism of several metabolic diseases, such as T2D, NAFDL and MetS. The discovery of ncRNAs and their involvement in insulin signaling pathway has unveiled new opportunities to elucidate the molecular mechanisms leading to IR and related diseases, in order to prevent disease onset and progression. Indeed, ncRNAs have emerged as reliable disease biomarkers and promising therapeutic targets, although further studies are needed before clinical translation into diagnostics and therapy.

## Figures and Tables

**Figure 1 ijms-22-07716-f001:**
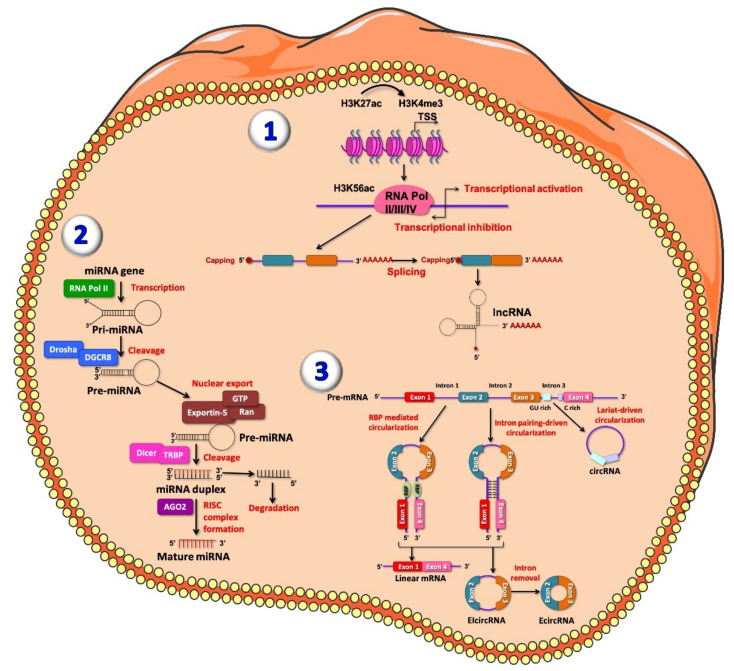
Representative figure of long non-coding RNAs (lncRNAs) (**1**), microRNAs (miRNAs) (**2**) and circular RNAs (circRNA) (**3**) biogenesis. (**1**) At the chromatin state, H3K27ac and H3K4me3 are enriched at lncRNA promoter; transcription of lncRNA is initiated from different promoters in antisense direction, enriched for H3K56ac and Pol II/III/IV. The resulting pre-mature lncRNA is subjected to a 3′-polyadenylated and the 5′-end capping with methyl-guanosine. Then, all introns are spliced, resulting in a final mature lncRNA. (**2**) MiRNAs are firstly transcribed by RNA polymerase II into the nucleus, producing primary miRNAs (pri-miRNAs), a stem loop shaped RNA sequence. Pri-miRNA, once processed, is recognized and cleaved by the multi-protein complex Microprocessor within the nucleus. This complex is composed by two double-stranded molecules: RNase III enzyme DROSHA and RNA-binding protein DGCR8. DROSHA cuts, by its RNase III domains, in two different points of the double strand RNA (dsRNA) towards the base of the stem-loop generating a ~70 nucleotide hairpin–shaped precursor miRNAs (pre-miRNAs), showing an overhang at the 3′ end of 2 nucleotide left by the asymmetrical cut made by DROSHA recognized by Exportin-5 which carries the pre-miRNA into the cytoplasm. Here, the pre-miRNA is further processed by DICER/TRBP complex, which generates imperfect duplexes of 22 nucleotides containing a guide strand and a passenger strand. The guide strand (represented in red) together with Argonaute proteins forms RNA-induced silencing complex (RISC) and generates the mature miRNA, while the passenger strand is finally degraded. (**3**) CircRNAs are generated by an alternative splicing mechanism of pre-mRNA, termed back-splicing. In this process, the 3′-end of an exon binds to the 5′-end of its own or to an upstream exon through a 3′,5′- phosphodiester bond, forming a closed structure with a back-splicing junction site. Two models of circRNAs biogenesis have been described: the lariat model and the direct back-splicing model, further subdivided into RBP-mediated circularization and Intron pairing-driven circularization, regulating adjacent splice sites [52]. Lariat-driven circularization occurs through the interaction between the 3′ hydroxyl of the upstream exon with the 5′ phosphate of the downstream exon generating a covalent binding, producing a lariat containing both exons and introns. From both RBP-mediated circularization and intron pairing-driven circularization four main subtypes of circRNAs have been identified: exonic-circRNAs (ecircRNAs), mainly derived from single or multiple exons and exonic-intronic circRNAs (EIciRNAs), which consist of both introns and exons.

**Figure 2 ijms-22-07716-f002:**
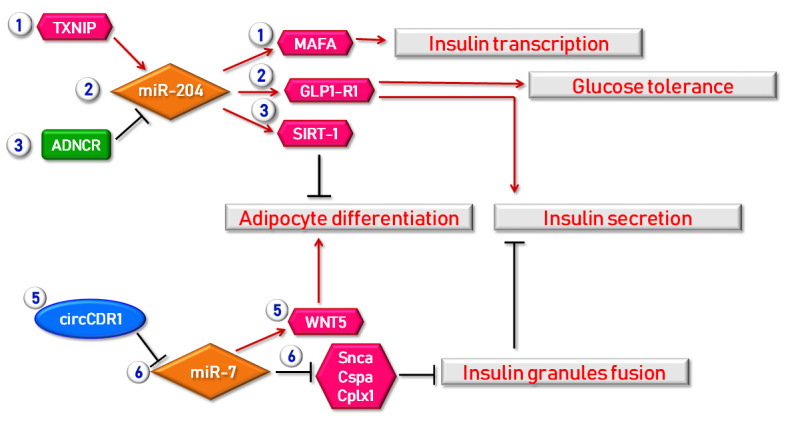
Examples of regulatory mechanisms in which lncRNAs, circRNAs and miRNAs are involved in obesity and β cell function. A first mechanism through which beta cell function is regulated by miRNAs is the TXNIP/miR-204/MAFA regulatory axis. Indeed, TXNIP (Thioredoxin-Interacting Protein) is able to induce miR-204 expression consequently leading to MAFA suppression and finally to insulin transcription inhibition. The same miR-204 also targets and regulates the expression of GLP1R (Glucagon-Like Peptide 1 Receptor) thus improving both insulin secretion and glucose tolerance. Moreover, miR-204 expression is also regulated by ADNCR (Adipocyte Differentiation-Associated lncRNA) lncRNA, thus leading to SIRT-1 overexpression, finally resulting in the repression of adipocytes differentiation. As a matter of fact, adipocyte differentiation is also regulated by circCDR1 acting as miRNA regulator for miR-7, in turn modulating WNT5 expression. Furthermore, miR-7 is also involved in insulin secretion modulation; indeed, this miRNA directly inhibits genes acting in insulin granules fusion (Snca, Cspa, Cplx1). Red arrows indicate enhancement/stimulation, while black lines represent inhibition/suppression.

**Table 1 ijms-22-07716-t001:** Summary of the cited lncRNAs involved in metabolic diseases and their mechanisms of action.

Name	Functions	Targets
SRA	co-activator of PPARγ2 in adipogenesis [102]; promotes steatosis, ↓FFAs oxidation [102]	PPARγ2 [102]; FoxO1 [102]
ASMER1, ASMER2	adipogenesis, lipolysis, adiponectin secretion [103]	-
ADNCR	adipocyte differentiation inhibition [104]	miRNA-204/SIRT1 [104]
H19	↓expression of adipogenic factors [102]; inhibition of BMMSCs adipogenic differentiation [105]; promotes hepatic lipid accumulation [111]; promotes gluconeogenesis [102,112]	HDACs 4,5,6 [102]; miR-675/HDACs [105]; PTBP1/SREBP-1c [111]; FoxO1 [102], HNF4a [112]
HOTAIR	promotes pre-adipocyte differentiation, ↑adipogenic genes expression (PPARγ and LPL) [101,106]; activates fibrogenesis and promotes TG accumulation [21]	Genes at HOXD locus [101]; MMP2, MMP9 [21]
BlnC1	↑UCP1 expression, ↑ mitochondiral genes expression [3]; promotes adipose tissue browning [102]	Ebf2 [3,101,102]
MALAT1	↑ROS and pro-inflammatory cytokines, promotes angiogenesis [112]; involved in the pathogenesis of diabetic gastroparesis [113]; promotes liver steatosis and fibrosis [21,102,114,115]	SAA3 [112]; α-SMA and SM-MHC [113]; CXCL5 [21,114,115]
ANRIL	promotes atherosclerosis, ECs proliferation and adhesion [116]; causes cardiac dysfunction, myocardial apoptosis and fibrosis [117]	INK4B [116]
MEG3	promotes ECs functions, ↑microvascular permeability and inflammation upon silencing [116]; fibrosis progression upon downregulation, possibile interaction with HOTAIR [21]	PI3K/Akt [116]
CASC2	key role in renal tumorigenesis, possible involvement in renal failure [118]	-
MIAT	↑TNFa and VEGF, regulates ECs function [116]	miR-150-5p/VEGF [116]
NONRATT021972	involved in nociception and neuropathic pain ↑TNFa [119]	TNFa [119]
lncARSR	↑hepatic lipid accumulation [120]	PI3K/Akt /SREBP-1c [120]
Lnc18q22.2	associated with fibrosis and lobular inflammation [121,122]	-
APTR	involved in collagen accumulation and HSCs activation [21]	-

FFAs: free fatty acids; BMMSCs: bone marrow mesenchymal stem cells; LPL: lipoprotein lipase; TG: triglycerides; ECs: endothelial cells; HSCs: hepatic stellate cells.

**Table 2 ijms-22-07716-t002:** Summary of the cited miRNAs involved in metabolic diseases and their mechanisms of action.

Name	Functions	Targets
miR-130	↓ adipogenesis [135]	PPARγ [135]
miR-486	↑preadipocyte proliferation and myotube glucose intolerance [137]	preadipocyte [137]
miR-146b and miR-15b	↓glucose-stimulated insulin secretion [137]	-
miR-375	↓GSIS process [138]	insulin gene transcription [138]
miR-7	↓expression of genes involved in the process of fusion of insulin granules with plasma membrane and SNARE proteins [97]	genes involved in insulin granules fusion with plasma membrane and SNARE proteins [97]
miR-184	↑ β-cell proliferation, ↑insulin secretion in response to glucose challenge and protects β-cells against apoptosis induced by chronic exposure to proinflammatory cytokines or FAs [139,140]	β-cells in pancreatic islets [139,140]
miR-24	implicated in de-differentiation and protection against apoptosis of β-cells [141]	β-cells in pancreatic islets [141]
miR-204	↓insulin production [142]; ↑ improved glucose tolerance and insulin secretion [143]	TXNIP/miR-204/MafA/insulin pathway [142]; GLP1R [143]
miR-122	↑Hepatocyte proliferation and maturation [144] ↓TG synthesis [145]	HNF6 [144]
miR-29a	↑Uptake of lipids from the circulation in hepatocytes [146]	Lpl [146]
miR-21	↓TG, free cholesterol, and total cholesterol levels [147] ↑Lipid oxidation [148]	HMGCR and FABP7 [147,149]; PTEN [150]; PPARα [148]
miR-33	Cholesterol and FAs metabolism [151]; ↓Gluconeogenesis [152,153,154,155]	ABCA1 and ABCG1, CPT1A and AMPKα [151]; PCK1, G6PC [152,153,154,155]
miR-34a	↓FAs catabolism [156]	PPARα, SIRT1 [156]

**Table 3 ijms-22-07716-t003:** Summary of the cited circRNAs involved in metabolic diseases and their mechanisms of action.

Name	Functions	Targets
circRNA_1897	↑Adipocytes differentiation and lipid metabolism [192]	miR-27a and miR-27b-3p [192]
circRNA_26852	Lipolysis and inhibition of adipocyte differentiation via a PPARγ-dependent mechanism [192]	miR-874 and miR-486 [192]
circ_0046367	Activation of genes involved in lipid metabolism such as Cpt2 and Acbd3 [193]	miR-34a/PPARα [193]
circHIPK3	↑Lipid droplets accumulation following oleate treatment [194]	miR-192-5p/Foxo1 [194]
circ_0001946(circCDR1)	↑Adipogenic differentiation ↓Osteogenic differentiation of BMSCs [195]; ↓Insulin secretion [197]	miR-7-5p/Wnt5 [195]; miR-7/Snca, Cspa, Cplx1 [197]
circRNA_0095570 (circH19)	↓Lipids droplets formation and adipogenesis in ACDCs cells	Pparγ, Srebp1c and Cebp1- α
circ_Arhgap5-2	↓Adipogenesis markers [196]	Pparγ, Cebpα, Fabp4 and AdipoQ [196]
circRNA_0054633	Biomarker of prediabetes in peripheral blood [198]	-
circ_ANKRD36	Potential role in inflammatory mechanisms occurring in T2D [199]	-
circRNA_021412	↓long chain acyl-CoA synthetases and development of hepatic steatosis [200]	miR-1972/LPN1 [200]
circRNA_0046366	↑Hepatocyte steatosis [193]	miR-34a/PPARα [193]

BMSCs: bone marrow stromal cells; ACDCs: adipose derived stem cells.

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
