# Peer review of "Non-Coding RNAs: Novel Players in Insulin Resistance and Related Diseases"

_ijms, 2021, doi:10.3390/ijms22147716_

Round 1

Reviewer 1 Report

Formichi et al. comprehensively reviewed the role of ncRNAs in insulin signaling. The wealth of information included is extensive. However, I have the following comments.

a) Authors should include a table of ncRNAs with their mechanisms, which will serve as a snapshot.

b) Could you go through the sentences carefully, some are too long with structural errors.

Author Response

Dear Editor,

On behalf of all the authors, I am submitting the revised manuscript entitled “Non-coding RNAs: novel players in insulin resistance and related diseases” to be evaluated for publication in International Journal of Molecular Sciences.

We thank the Referees for their useful comments.

We included a table of ncRNAs discussed in the manuscript with their mechanisms of action. We revised the manuscript to emend English errors and rephrase/rewrite sentences with high similarity index. Further, we revised longer sentences, as suggested.

Reviewer 2 Report

  1. Review content – In its current form, this manuscript provides intensive information about non-coding RNA involved in insulin resistance. It would be much informative if the authors further provide any therapeutic strategies targeting non-coding RNA for the treatment of insulin resistance and related diseases. Also, it would be much informative if the authors summarize different MoA among non-coding RNAs.
  2. Other broader questions that a synthesis of the different examples include in conclusions could be: what are the relative MoA contributions of different modality of drugs targeting non-coding RNA insulin resistance related disease, such as T2D? Which signaling pathway appear to be more frequently preferred? Any general patterns of drug are used in different therapies?

Author Response

Dear Editor,

On behalf of all the authors, I am submitting the revised manuscript entitled “Non-coding RNAs: novel players in insulin resistance and related diseases” to be evaluated for publication in International Journal of Molecular Sciences.

We thank the Referees for their useful comments.

We added some information on ncRNA-based therapeutic strategies. Data on such treatment option are quite scarce and ncRNA-based therapy are still not available for metabolic syndrome management. However, ncRNAs mimics or antagonists are being studied to address peculiar molecular pathways responsible for metabolic diseases onset, such as inflammation, angiogenesis and fibrosis. NcRNAs’ mechanisms of action  have been summarized in Table 1 -3 in the manuscript.